# Does Insect Aversion Lead to Increased Household Pesticide Use?

**DOI:** 10.3390/insects13060555

**Published:** 2022-06-18

**Authors:** Tali Leibovich-Raveh, Moshe Gish

**Affiliations:** 1Department of Mathematics Education, Faculty of Education, University of Haifa, Haifa 3498838, Israel; talil@edu.haifa.ac.il; 2Department of Natural Resources and Environmental Management, Faculty of Social Sciences, University of Haifa, Haifa 3498838, Israel

**Keywords:** entomophobia, fear of insects, insecticides, insectophobia, IPM, pest control, pesticide exposure, pesticide poisoning, urban pests

## Abstract

**Simple Summary:**

Domestic pesticides are commonly used for killing insect pests such as cockroaches and ants, despite the potential health risks associated with pesticide exposure, especially for young children. If feelings of fear and disgust toward insect pests are motivators that can encourage people to use more pesticides than they should, it may be worthwhile to try and mitigate such feelings in both children and adults, in order to lower current and future unnecessary exposure to domestic pesticides. Since there are no studies on the possible link between the common aversion to insects and domestic pesticide use, we tested the level of aversion and recorded pesticide use among volunteers. Surprisingly, we did not find such a connection, possibly due to other, more dominant factors that affect peoples’ decisions. We did, however, identify several other attributes such as fear of toxic chemicals, vegetarianism and infestation levels, which clearly affected pesticide use patterns. Our findings may be culture-specific and relevant to the population we sampled, but future studies that should be conducted in other societies may identify different motivations for using pesticides. Such insights may improve the efforts of public health authorities to lower the non-dietary exposure to pesticides in the home environment.

**Abstract:**

In many human societies, domestic insect pests often evoke feelings of disgust, fear and aversion. These common feelings may translate to increased use of household pesticides. No study has ever explored this possibility and consequently, efforts to mitigate public exposure to domestic pesticides typically focus on addressing knowledge gaps. We tested the hypothesis that negative emotions toward insects may motivate people to use pesticides, by interviewing 70 participants and assessing their insect aversion levels using a computerized test. Contrary to our hypothesis, we found no effect of insect aversion on pesticide use. However, we did find that personal attributes and preferences such as wishing to avoid exposure to toxic chemicals, being vegetarian and taking frequent nature walks reduced pesticide use, in addition to low infestation levels and physical attributes of the housing unit. We emphasize the importance of conducting future studies in various societies, where insect aversion and other factors may have different effects on household pesticide use. Such studies may provide culture-specific insights that could foster the development of next-generation urban IPM (Integrated Pest Management) public education programs, which will address not only knowledge gaps, but also emotional aspects and personal attributes that lead to unnecessary or excessive use of household pesticides.

## 1. Introduction

Household pesticides are used worldwide for the preventive and responsive control of domestic pests. Whether applied by household members or professional exterminators, pesticides provide convenient and cost-effective solutions to domestic pest problems. Up until the end of the 20th century, organophosphate pesticides dominated the residential chemical pest control market in the USA [1]. Due to their adverse effects on human and environmental health [2], by the end of 2005, the United States Environmental Protection Agency completed a phase-out of organophosphate pesticides from residential use [3] and consequently, the market shifted to pesticides with low toxicity to humans [2,3,4,5]. Today, most household pesticide products (henceforth, HPPs) in the USA are based on pyrethroids (synthetic versions of pyrethrins—botanical insecticides) [6,7], which have become the most widely used class of insecticides in the world [8].

Although pyrethroids are considered relatively safe to humans due to their low mammalian toxicity [2,8,9], a growing body of evidence raises concerns about the safety of pyrethroids [10,11,12,13,14,15,16,17] and adjuvants [18] contained in HPPs. Therefore, modern HPPs are not entirely safe, especially since they are mostly sold as high-exposure applications such as spray cans and pest bombs [5] and since pyrethroids are relatively stable and persistent in indoor environments [19,20,21]. Moreover, pesticides tend to settle on indoor surfaces and in dust that accumulates both in and around the house [22,23,24,25,26,27,28]. Even though it is difficult to directly monitor household pesticide use [29], numerous human biomonitoring studies have indeed found a ubiquitous, non-occupational exposure of the general public to pyrethroids, through food contamination and household pesticide use [13,30,31,32,33,34,35,36].

The practice of limiting the public’s dietary exposure to pesticides is relatively straightforward, since farmers are normally subjected to laws and regulations regarding the safe use of pesticides, and produce can be easily inspected for pesticide residues by health authorities and private laboratories. However, non-dietary exposure to pesticides in the home environment is inherently difficult to control, since it depends on behaviors and decisions made by the general public, who often lack appropriate knowledge or awareness and tend not to consult pest management professionals [37]. In addition, HPPs are sold freely in retail shops and after they are purchased, health authorities have practically no control over the manner of their use. Therefore, the success of efforts to reduce non-dietary pesticide exposure in homes depends on a currently lacking understanding of the circumstances that lead people to use pesticides the way they do, especially those that result in excessive, unnecessary or irresponsible use.

Different factors may lead to increased non-dietary exposure of household members to pesticides, including: the severity of a pest problem, advertisements for HPPs and extermination services, high availability and low prices of pesticide treatments, ease of use of HPPs, careless reading of labels, lack of awareness of risks [5,38] and the perception of modern pesticides as “green” or less problematic than the pests they are meant to control [39,40]. Initiatives aimed to reduce the public’s exposure to household pesticides often address these factors by providing accessible information such as explanations on pests, integrated pest management practice and the safe use of pesticides in and around homes e.g., [9,41,42,43,44], or by providing relevant guidance to government authorities and other stakeholders [44].

Arguably, communicating information to the public targets only the part of the problem that can be modified by rational thinking and awareness. However, the emotional component of household pesticide use is not usually considered in public education programs, even though the motivation to use pesticides may be driven not only by knowledge and awareness, but also by emotions, perceptions and attitudes toward pests and pesticides [37,45]. Studies in developed countries indeed found correlations between self-reported attitudes toward arthropod pests and the use of household pesticides [45,46,47,48,49]. However, no study, to the best of our knowledge, has examined the effects of feelings such as fear, anxiety and disgust towards arthropods (hereafter referred to as “insect aversion”, even though these feelings may often be equally relevant to other groups of arthropods) on decision making that leads to indoor pesticide use (an exception is the extreme, rare case of delusory parasitosis (Ekbom syndrome—a delusional condition in which a person believes they are infested by invisible bugs)), which often leads to the application of large and sometimes dangerous amounts of pesticides [50,51]. In other words, it is currently unknown whether insect aversion can affect quantities and frequencies of household pesticide use.

Since sociological and psychological conceptions can affect peoples’ decisions on how to manage household pests [37], it is reasonable to assume that a higher level of personal insect aversion can often lead to a higher frequency and intensity of pesticide use. However, similar negative attitudes and feelings toward chemicals may counter this effect [52,53]. Thus, in order to develop new strategies for mitigating household pesticide use that will address emotional motivators (for example, promoting greater tolerance to insects among children and adults), the current void in understanding how different factors shape the decision-making process that leads to pesticide use must be filled. Our study therefore aimed to begin addressing this knowledge gap by assessing participants’ level of insect aversion, pesticide use, level of indoor exposure to insects and their predisposition to solving pest problems with pesticides. The study was conducted on a specific and relatively homogenous population, as explained in Section 2.2.

## 2. Materials and Methods

### 2.1. Study Area

Our study was conducted in northern Israel and designed with consideration of several characteristics of domestic insect pest infestations in Israel: (1) The dominant household pests in Israel are cockroaches (mostly *Periplaneta americana* and *Supella longipalpa*) and ants of various species (bed bugs are very rare in Israel). (2) Houses in Israel are typically built out of bricks and concrete, therefore, cryptic wood infesting insects are negligible as domestic pests. In addition, homes in Israel usually do not have basements, which in other countries tend to harbor a wide and unique diversity of arthropods adapted to cave-like environments [54,55].

### 2.2. Participants

We collected data from 70 secular Jewish families, in August–October 2019. We visited families in their homes and interviewed and tested only the mothers (23–63 years old), to avoid bias by the possible tendency of men to hide feelings of fear and disgust [56,57]. We preferred interviewing the participants in person rather than conducting an online study since remote questionnaires for self-reporting of pesticide use may be less reliable than interviews [40]. All participating women lived with a male partner and at least one minor child or at least one adult child that visits at least once a month. All participants signed a consent form after volunteering for the study. The study was approved by the ethics committee of the University of Haifa.

### 2.3. Procedure

We collected two types of data: (1) The degree of insect aversion of the participant—this was evaluated using a computerized test (described below); (2) Estimated indoor exposure to insects, factors affecting pesticide use, and the use of pesticides in the household. These data were collected using a questionnaire (described below) and checking and listing all HPPs that were present in the house at the time of the researcher’s visit (for a detailed list of all HPPs found in the participants’ homes, see Appendix A). Spot-on products and on-animal sprays for controlling pet ectoparasites were not considered as HPPs in this study, since they are not applied to indoor surfaces and the reasons for using them are not related to most factors tested in this study.

The exact focus of the study was revealed to the participants only after they had completed the computerized test, which works best when participants are unprepared for what they are about to experience during the test. Therefore, participants were initially told that the study subject was “consumption habits and risk factors in the home environment”. The order of actions was as follows: computerized test, questionnaire, revealing of the exact study subject, checking what HPPs there were in the house.

### 2.4. Computerized Test

Participants performed the computerized test on a laptop with a Windows 10 operating system while sitting in a quiet room, to avoid distractions. The test was designed to provide a quantitative estimate of the participants’ insect aversion level. The computerized test was programmed and carried out using OpenSesame 3.1 [58] software. The test was composed of 20 trials, using 20 different, randomly ordered pictures and lasted less than 10 min. The pictures we used were of hands holding cockroaches, which on top of being very common household pests, were found in several studies to be the least liked arthropod species [49,59]. We chose pictures of physical contact with a disgusting stimulus in order to amplify the negative feelings toward it [60,61].

In each trial, a picture was presented on the screen for 800 milliseconds. Participants then had to grade each picture according to the level of unpleasantness it made them feel (Figure 1). A fixed exposure time to the stimulus prevented variation in responses that could have been created by differences in exposure times.

### 2.5. Questionnaire

After the computerized test has ended, the researcher interviewed the participant using a questionnaire composed of three sections, designed to assess three different characteristics: (a) the level of indoor exposure to insects (henceforth “exposure”); (b) the extent of pesticide use (henceforth “pesticide use”); (c) factors that may affect the family’s predisposition toward using pesticides in their home (henceforth “tendency”). To the best of our knowledge, there is no validated questionnaire in the literature designed to quantify these three characteristics of a household and household members and, therefore, we composed a questionnaire that we consider to be exploratory. In Table 1, we provide an explanation of the reasoning behind each question in the questionnaire. For each question, each possible answer contributed a certain number of points to a total grade in that section (on a scale of 0–12), which was the index score for that characteristic.

#### 2.5.1. Questionnaire Section A: Exposure

In this study, we chose to assess the subjective opinion of the participant on infestation levels in her home, rather than to perform an objective, trap-based monitoring of pest population size. The reason for this was that it is the subjective experience of people that motivates them to use pesticides, not necessarily the real infestation levels, which they may not be aware of [62]. Therefore, we based the assessment of the level of indoor exposure to insects on structural attributes of the apartment/house and on subjective reporting by the participant (Table 1, section a).

#### 2.5.2. Questionnaire Section B: Pesticide Use

The intensity of pesticide use in the household was estimated using four questions and an inventory list of HPPs present in the house at the time of the researcher’s visit (Table 1 section b).

#### 2.5.3. Questionnaire Section C: Tendency

We defined “tendency” as the predisposition of a family to using pesticides, which derives from characteristics that are not emotional aversion to insects. In this section, we used four factors to assess the family’s tendency to use pesticides (Table 1, section c).

### 2.6. Data Analysis

We used a stepwise multiple linear regression analysis to predict pesticide use based on tendency, exposure, and the level of insect aversion of the participant. We examined the relationship between insect aversion and pesticide use using a Pearson correlation test. We also evaluated the strength of the evidence using Bayesian statistics, which is less affected by sample size. Specifically, we report the Bayes Factors (BF), which express the ratio between the evidence in favor of the experimental hypothesis relative to the null hypothesis. BF with a value that is close to 1 indicate that the analysis is not sensitive enough and more data should be collected. A BF value of over 3 suggests that the analysis is sensitive enough for accepting the experimental hypothesis [63]. We defined the BF as a ‘null’/experimental hypothesis, where ‘null’ suggests no correlation between the factors. All statistical analyses were performed using JASP software [64].

## 3. Results

The descriptive statistics of the data collected in the study are given in Figure 2.

Before conducting the stepwise multiple linear regression analysis with the factors tested in this study (i.e., tendency, exposure and insect aversion), we preformed correlation tests to check whether correlations between different factors exist and if so, can they result in multicollinearity. There was no correlation between any of the predictors (Table 2). Therefore, we used all factors in the stepwise linear regression for the prediction of insecticide use based on insect aversion, tendency and exposure.

The results of the regression revealed two regression models. In the first model, tendency was the only predictor, explaining 20.5% of the variance (R = 0.45, F (1.69) = 17.54, *p* < 0.001). In the second model, both tendency and exposure together explained 37.2% of the variance (R = 0.61, F (2.69) = 19.84, *p* < 0.001). According to the second model, tendency predicted insecticide use (ß = 0.522, *p* < 0.001), as did exposure (ß = 0.415, *p* < 0.001). Insect aversion was not included in the model and was therefore not a predictor (for details, see Table 3).

We further explored the lack of effect of insect aversion on pesticide use found in the stepwise regression analysis by correlating insect aversion with pesticide use. This analysis resulted in a null effect (Pearson’s r = 0.03, *p* = 0.804). The BF of the correlation between insect aversion and pesticide use was 6.5. Hence, the null hypothesis is 6.5 times more likely than the alternative hypothesis, suggesting that indeed, insect aversion did not predict pesticide use and this null effect was not due to lack of statistical power.

## 4. Discussion

Feelings of aversion toward invertebrates are common in western societies [65,66,67,68], yet proper characterization and quantification of this phenomenon is lacking. Insect aversion may be affected by multiple factors [69], but it may also affect different aspects in a person’s life. One possible outcome of high insect aversion may be an increased use of pesticides in the home environment. Despite the potential contribution of insect aversion to the public’s exposure to pesticides, to date, this connection is still hypothetical. Our study began with an intuitive hypothesis—that the greater the feelings of disgust or fear a person feels toward insects, the higher the frequency and amounts of pesticides they will use in their homes. The results of our study did not support this hypothesis. Assuming that decisions regarding pesticide use are not arbitrary, there may be at least three explanations for this finding: (1) Such a connection between insect aversion and pesticide use does exist in the population we sampled, but the methodology we used was not optimal for detecting it (for example, the weights of the factors in Table 1 should be different than the ones we have presented). (2) Insect aversion has some effect on pesticide use, but it is masked or counteracted by other factors, either the ones we tested or others that were not included in this study. (3) There is no effect of insect aversion on household pesticide use in the population we sampled, meaning that other factors determine pesticide use patterns. Indeed, we found the two other factors included in this study—exposure and tendency—to be predictors of pesticide use. In the population we sampled, insect aversion may have little or no contribution to the dominant effect of these two factors.

Our finding that the level of indoor exposure to insects affects pesticide use is not new [5], and it is rather expected: the greater the frequency of insect sightings, the greater the need people feel for taking action against the pests. While exposure level is relatively simple to assess, tendency cannot be measured directly and therefore we estimated it using several characteristics that we assumed to be indicators of a family’s tendency to use pesticides in their home (Table 1, section c). The most important characteristic in this estimate (biggest contribution to the total score in this section of the questionnaire) was wishing to minimize the exposure of household members to toxic chemicals. Our reasoning was that the stronger the motivation to avoid toxic chemicals, or “chemicals” in general (chemophobia [52,53]), the more dominant will it be over insect aversion, in determining pesticide use patterns. The second characteristic we used to estimate tendency was vegetarianism of household members. Since the most common motivation for being vegetarian is the concern about killing animals and animal welfare [70,71], vegetarians may tend to seek alternative solutions to pest problems that do not include the use of pesticides for killing them. The third characteristic in the tendency assessment was having pets at home. The presence of pets, in itself, does not affect indoor insect diversity and community composition [55]. Pets should also have a negligible effect on contamination levels of the home environment with pesticides, since the most effective, and therefore popular, modern treatments for pet ectoparasites are spot-on products, on-animal sprays and pills [72]. However, since pesticides contained in HPPs (especially pyrethroids) are a major cause of poisoning in domestic pets such as cats and dogs [73,74], we assumed that pet owners may be more cautious when it comes to storing and using pesticides in their homes. Moreover, owning pets may be an indicator of more positive attitudes toward animals, including unpopular ones such as ants [75]. The fourth characteristic was the frequency of nature walks, which indicates the level of exposure to nature. People who are more exposed to nature or enjoy nature-related leisure activities, may have higher tolerance to invertebrates [66,76]. Consequently, those people may feel a lesser need to kill insects they encounter in their homes (i.e., may prefer to trap the insect and throw it outside). In Israel (where this study was conducted), urban centers are not very big and as a result, natural areas in which people can take nature walks are accessible even for urban residents. Future studies in other countries, which will include residents of large urban areas with little access to natural areas, may need to ask participants about alternative activities that may also provide an experience of nature, such as gardening and spending time in urban parks [77]. Overall, our findings indicate that the personal characteristics that contributed to what we defined as “tendency” (which include the frequency of nature walks) can shape peoples’ behavior related to how they use pesticides in their homes.

It is interesting that in the population we sampled, insect aversion did not have a significant effect on household pesticide use. It is likely that in other societies, differences in insect aversion levels, sanitation standards, education levels, awareness to health risks of pesticides and exposure to HPP advertising will translate to differences in household pesticide use and the extent to which insect aversion affects it. For example, in societies where entomophagy is common practice [68,78], a passing cockroach may become a nutritious snack, whereas in other societies, it may end up being sprayed with a pesticide. Even among western societies, where entomophagy is rare [68,78], the average level of insect aversion may vary significantly due to differences in cultural conventions. It is possible that in the population we sampled, the level of insect aversion was relatively low, and that in other societies with higher levels of insect aversion, “tendency” factors and exposure levels will be overwhelmed by feelings of aversion, leading to excessive use of pesticides in homes.

It is therefore important that the design of future measures for mitigating the exposure to household pesticides in different societies, will consider emotional motivators and other factors that may affect pesticide use. Future studies that could facilitate such efforts might need to adjust or improve current methods for assessing insect aversion.

One type of action that may already be contributing to public health by suppressing the emotional motivation to use pesticides is the positive exposure of children and adults to insects. Today, entomology departments, nature museums, zoos, etc., around the world hold “insect festivals”, meant to alleviate fears, educate and address misinformation, and improve the overall attitude toward insects and other critters [79]. It may very well be that by reducing insect aversion among children today, we can lower the unnecessary exposure of their future family members to household pesticides.

Moreover, the mere exposure of children and adults who live in urban areas to general nature experiences may have a long-lasting, suppressing effect on insect aversion [80]. It is widely accepted that increasing opportunities to experience nature among urban residents can provide several benefits, such as raising awareness to the importance of biodiversity conservation [81] and promoting human health and well-being [82]. We suggest another way in which combating the “extinction of experience” [77] may improve human health in some societies: a higher tolerance to insects, which may lead to decreased pesticide use. However, as intuitive as this idea might seem, it is not an outcome of this study, nor is it currently supported by any other study we are aware of. We believe that future studies on this topic that should be conducted in various countries may contribute to the growing understanding of the importance of exposing children and adults to nature.

## Figures and Tables

**Figure 1 insects-13-00555-f001:**
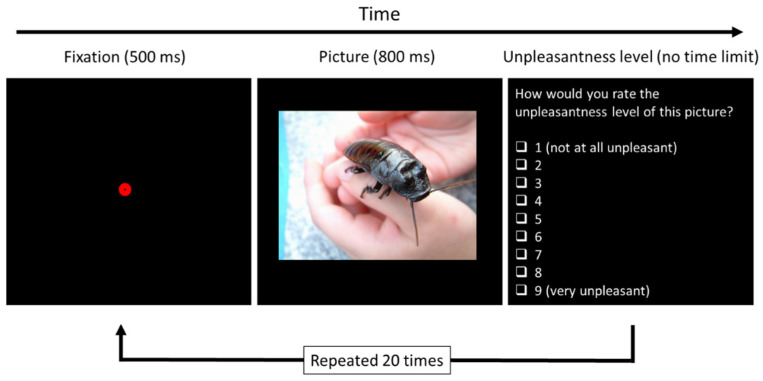
The design of the computerized test for determining insect aversion level. A fixation dot was presented on the screen for half a second, followed by a picture of a hand holding a cockroach. The participant was then asked to rate how unpleasant the picture made her feel (ms = milliseconds). Photo credit: Kevin Rohr (freeimages.com; accessed on 8 May 2022).

**Figure 2 insects-13-00555-f002:**
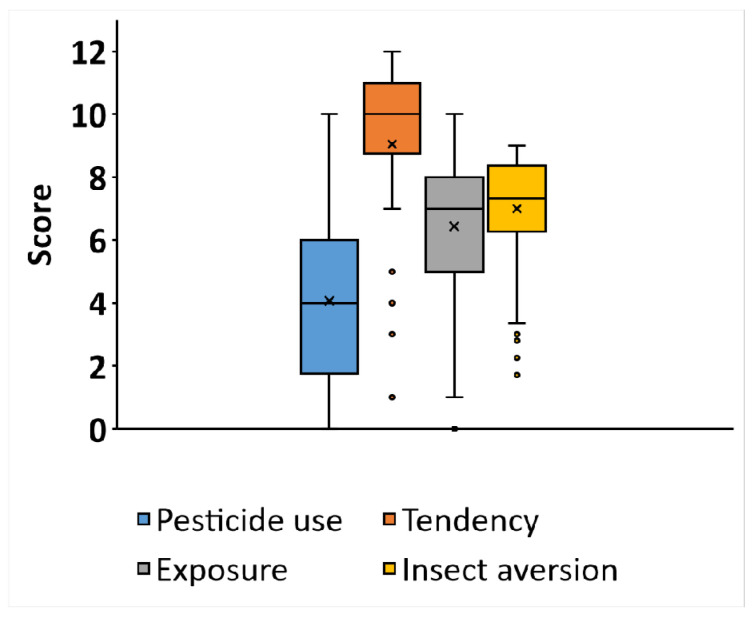
Descriptive statistics (box plots) of the data from the computerized test for insect aversion and from the questionnaire (*n* = 70). Each box represents a different section in the questionnaire. The X mark represents the mean, the line is the median, box borders represent the interquartile range (25th–75th percentiles), and the whiskers represent minimal and maximal values in the range of 1.5 IQR. Dots represent outliers that exceed a distance of 1.5 IQR.

**Table 1 insects-13-00555-t001:** An English translation of the questionnaire used in the study (originally in Hebrew). (**a**) Questions and scores used to assess the potential indoor exposure to insects. The range of the total score was between −3 and 9. Three points were added to the total score, thus creating a value between 0 and 12. A higher total score in this section indicated a higher potential for indoor encounters with insects and a higher perceived infestation level. (**b**) Questions and scores used to assess the level of pesticide use in the house. The maximum number of HPPs that were documented in a single household was 3, therefore, the range of the total score in this section was 0–12. A higher total score in this section indicated a higher level of pesticide use. (**c**) Questions and scores used for estimating the family’s tendency to use pesticides. The range of the total score was between −7 and 5. Seven points were added to this score (the corrected value was between 0 and 12). A higher total score in this section indicated a higher tendency to use pesticides.

Questionnaire Section	Question	Reasoning	Possible Responses	Score
a (Exposure—the level of indoor exposure to insects)	Type of home	Ground level rooms tend to have a higher diversity of insects than higher floors [1].	Private house/ground floor	3
			An apartment on the 2nd floor or higher	0
	Are there screens on the windows?	Screens prevent insects from entering through windows.	No	2
			Yes, on some windows	1
			Yes, on all windows	0
	During spring and summer, how frequently do you see insects inside your home?	The frequency of insect sightings is a major component of a person’s perception of infestation levels.	Very high frequency	4
			High frequency	3
			Medium frequency	2
			Low frequency	1
			Almost never	0
	If you do not use pesticides, what is the reason? *	No need to use pesticides indicates a low infestation level.	No need	−3
			Irrelevant (because there is insecticide use)	0
b (Pesticide use—the extent of pesticide use in the home)	If pesticides are used in your home, who does the extermination?	Households that add professional extermination to HPP use likely have an overall higher level of pesticide use.	Irrelevant (because there is no pesticide use)	0
			Family members	1
			Professional exterminator	1
			Both	2
	If family members do the extermination, what is the frequency of treatments?	Frequency of HPP use directly affects the level of pesticide use.	Irrelevant (or no more than once a year)	0
			Once in several months	1
			Once a month or more frequent	2
	If extermination is done by a professional exterminator, what is the frequency of treatments?	Frequency of professional extermination directly affects the level of pesticide use.	Irrelevant (no professional extermination)	0
			Once every two years	1
			Once a year or more frequently	2
	Is extermination usually prophylactic (against insects in general) or aimed at specific, existing pests?	General, prophylactic spraying “against insects” tends to be more widespread than responsive treatment of specific targets.	Irrelevant (no extermination)	0
			Specific	1
			General	3
	Number of pesticide products currently at home.	A higher number of HPPs indicates more use.	Number of products	1 point per product
c (Tendency—factors that may affect the family’s predisposition toward using pesticides in their home)	If you do not use pesticides, what is the reason?	Awareness of the toxicity of pesticides and preferring to avoid exposure to them should be negatively correlated with tendency (see discussion).	To avoid exposure to toxic chemicals **	−6
			Irrelevant (because there is pesticide use)	0
	Are any of the household members vegetarian?	Vegetarians may be less willing to kill pests (see discussion).	Yes	−1
			No	1
	How many pets do you keep at home, and of what kind (only mammals and birds)?	See discussion	None	2
			Pet/s of only one kind	1
			Pets of more than one kind	0
	How often do you take nature walks?	See discussion	Often (more than once a month)	0
			Once every month or two	1
			Once every three months or less	2

* This question appears twice—here and in section c in this table, since the first answer to this question—“no need”—is relevant here (infestation level) and the second answer—“to avoid exposure to toxic chemicals”—is relevant to the third section of the questionnaire regarding factors that may affect the tendency to use pesticides. ** Participants were free to name other reasons, but this was the only one given.

**Table 2 insects-13-00555-t002:** Correlation matrix for the different predictors tested in the study.

Predictor I	Predictor II	Pearson’s r	*p*
tendency	exposure	−0.168	0.164
tendency	insect aversion	0.034	0.779
exposure	Insect aversion	0.166	0.17

**Table 3 insects-13-00555-t003:** Stepwise regression results.

Model		Unstandardized ß	Standard Error	Standardized ß	*t*	*p*
1	(Intercept)	−0.157	0.317		−0.15	0.882
	tendency	0.467	0.111	0.453	4.188	<0.001
2	(Intercept)	−3.72	1.263		−2.945	0.004
	tendency	0.539	0.101	0.522	5.32	<0.001
	exposure	0.452	0.107	0.415	4.22	<0.001

## Data Availability

The data presented in this study are available on request from the corresponding author. The data are not publicly available due to private information contained within the interviews, which were conducted in Hebrew.

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
