# Peer review of "Does Insect Aversion Lead to Increased Household Pesticide Use?"

_insects, 2022, doi:10.3390/insects13060555_

Round 1
Reviewer 1 Report
The manuscript is well-written and well presented. Although the manuscript is not a typical scientific experimental research with a typical research methodology, it is still an important study to explore and improve our understanding of the public behaviors, personal attributes, and emotional motivators regarding household insecticide usage. The study seems to be conducted on certain populations, e.g., only the mothers or females, and if appropriate, this piece of information should be mentioned before the methodology section, for clarity. Additionally, in the introduction, the authors may add more sentences to emphasize the impact of this study on future designs and implementations of sustainable IPM strategies in urban environments and/or outreach programs that may lead to environmentally-responsible behaviors, subsequently, broader environmental benefits.
Author Response
Reviewer comment: “The study seems to be conducted on certain populations…”
Answer: We added a sentence that clarifies this at the end of the introduction (lines 111-112 of the revised manuscript).
Reviewer comment: “in the introduction, the authors may add more sentences to emphasize…”
Answer: We added a sentence to the introduction (lines 106-107 of the revised manuscript), but we feel that the best place to elaborate about this is in the discussion. We added a paragraph at the end of the discussion that addresses this suggestion of the reviewer.
Reviewer 2 Report
This is a very simple but impressive approach to an international problem from an entirely different perspective. And the suggestions to improve it and repeat in different societies are very important. Israel was a good place to conduct the first studies because of the relatively uniform climate and housing and the limited number of arthropod pests. The discussion on the reasons for limited pesticide usage were also on target, especially the mention of children being acclimated to insects by entomological societies. A nice paper to read and hope more will follow. The only suggest in the text is too much use of 'in order to' and the word 'aim' which seems to be widely used now. Suggested changes in the text.

Author Response
Answer: “in order to” was changed to other phrases in lines 201, 209, 225 and 245 of the revised manuscript.
Suggested changes in the text (line numbers refer to the original manuscript):
Line 143: change “to” to “for”. Corrected.
Line 150: change “aimed” to “was designed”. Corrected.
Line 169: change “aimed” to “designed”. Corrected.
Lone 173: change “aimed” to “designed”. Corrected.
Line 181: change “here” to “her”. Corrected.
Line 196: change “aimed” to “used”. Corrected.
Line 226: change “is” to “are”. Corrected.
Reviewer 3 Report
The author's efforts to understand the social aspects of urban pests and their management are interesting.
A clear conclusion of the study is needed.
This MS is more closely connected with social issues rather than technical/entomological issues.
Author Response
Answer: We made a few changes at the end of the discussion and added another paragraph. We believe that now, the last part of the discussion emphasizes more clearly what we did find and what we did not find in this study. Our main conclusion is that insect aversion did not affect pesticide use in the population we sampled, but other factors such as exposure to nature and infestation levels- did. We argue that this outcome may be specific to our study, and that it is important to conduct similar studies in other societies. We believe that now, this is better explained in the last part of the discussion.
Round 2
Reviewer 3 Report
The results and discussion sections could be presented better.